# Electrochemical, Biological, and Technological Properties of Anodized Titanium for Color Coded Implants

**DOI:** 10.3390/ma16020632

**Published:** 2023-01-09

**Authors:** Josef Hlinka, Kamila Dostalova, Kristina Cabanova, Roman Madeja, Karel Frydrysek, Jan Koutecky, Zuzana Rybkova, Katerina Malachova, Osamu Umezawa

**Affiliations:** 1Faculty of Materials and Technology, Department of Materials Engineering, VSB-Technical University of Ostrava, 17. Listopadu 2172/15, 708 00 Ostrava, Czech Republic; 2Centre for Advanced Innovation Technologies, VSB-Technical University of Ostrava, 17. Listopadu 2172/15, 708 00 Ostrava, Czech Republic; 3Trauma Center, University Hospital Ostrava, 17. Listopadu 1790, 708 52 Ostrava, Czech Republic; 4Institute of Emergency Medicine, University of Ostrava, Syllabova 19, 703 00 Ostrava, Czech Republic; 5Medin a.s., Vlachovicka 619, 592 31 Nove Mesto na Morave, Czech Republic; 6Faculty of Engineering, Yokohama National University, 79-5 Tokiwadai, Hodogoaya, Yokohama 240-8501, Japan

**Keywords:** titanium, anodization, corrosion properties, polarization, biocompatibility

## Abstract

Anodization coloring of titanium tools or implants is one of the common methods for the differentiation of each application by its size or type. Commercial purity titanium grade 4 plates (50 × 20 × 0.1 mm) were tested to obtain their electrochemical and other technological properties. The coloring process was done using the potential of 15, 30, 45, 60, and 75 Volts for 5 s in 1 wt. % citric acid in demineralized water solution. Organic acids solutions generally produce better surface quality compared to inorganic acids. The contact angle of colored surfaces was measured by the sessile drop method. Electrochemical impedance spectroscopy and potentiodynamic polarization were used for the determination of selected electrochemical and corrosion parameters of the tested surfaces. It was found that the anodization process decreases corrosion potential significantly. It was also confirmed that a higher potential used for anodization results in higher polarization resistance but also a decrease in corrosion potential. The anodization process at 75 V produces surfaces with the lowest corrosion rate under 1 nm/year and the noblest corrosion potential. It was confirmed that the anodization process in citric acid does not affect titanium cytotoxicity.

## 1. Introduction

During the last few decades, commercial purity (CP) titanium or titanium alloys became very popular in medicine and other fields where the combination of good mechanical properties, low weight, and high corrosion resistance is needed [1]. As there is a large variety of titanium implants and tools used in medicine, especially in implantology, there is a need for the quick differentiation between them as they can be similarly sized or shaped. Each tool or application can be simply marked by a different color, which allows one to distinguish the proper tool quickly [2]. As the electrochemical coloring process of titanium is very simple and no high-tech instruments are needed, it can be easily used in implant postproduction processes or as one of steps in implants manufacturing [3,4].

On the other hand, there are strict requirements on medical implants or other applications with regard to their corrosion properties as they are closely connected to ion release into surrounding tissue during their lifetime. Localized corrosion damage of anodically colored surfaces can also act as a stress concentrator, where cracks can nucleate [5]. The effect of bacterial implants’ surface colonization is not negligible and can be influenced by surface treatment processes [6]. The high surface roughness and wettability increases not only the cell adhesion, but also equally the risk of bacterial adhesion colonization, resulting in secondary severe complications e.g., periodontotis [7] or peri-implantitis [8]. Metabolic products of bacterial activity can also accelerate corrosion processes, causing intensive degradation of implants’ surfaces [9]. The oxide layer produced by electrochemical anodic oxidation in solutions of organic acids increases the surface corrosion resistance while changing the surface color, which is commonly used for implant color coding [2]. The anodization in inorganic acids generally produces less color variety and the homogeneity of the produced surfaces is also decreased due to significant surface blistering [10,11].

During the electrochemical coloring process, the character of the protective passive layer on the surface is significantly changed, especially in the case of titanium. The microstructure of the layer actively reacts with photon light when a photon of each wavelength can be absorbed or reflected [4]. The final color of the titanium surface is determined by the particular interaction between the oxide layer and light, and can be judged easily by comparative method using standards or by reflectometry methods [12]. The process of anodization is driven only by an external electric field, therefore, a slight change of anodization voltage can highly affect the shade or color of the anodized surface [13]. Variation of surface layer characteristics affects surface energy, which can result in a change of the surface contact angle. Electrical parameters of the layer can also be affected by this process. The colors formed on the titanium surface after anodizing are known as interference colors. There is no pigment associated with the production of these colored surfaces [14].

## 2. Materials and Methods

### 2.1. Subtrate Material

CP titanium grade 4 (chemical composition within the ASTM F67, Fe > 0.5%, C > 0.08%, O > 0.4%, N > 0.05%, and H > 0.015%) was used in this study, as this grade is commonly used in implantology for its valuable combination of mechanical and corrosion properties [15,16]. A large sheet of 0.1 mm thickness, supplied by Bibus Metal, was cut into smaller 50 × 20 mm pieces, which were rinsed by demineralized water and cleaned by acetone in an ultrasonic bath.

### 2.2. Anodization Procedure

For the preparation of the anodization solution, citric acid (C_6_H_8_O_7_, Sigma-Aldrich, St. Louis, MO, USA) was used. There was a 1% solution of citric acid in water prepared for the anodization solution. After each sample was marked, it was connected as the working electrode (anode) to a direct current supply (“Matrix MPS-7162”) and immersed into the solution electromagnetically stirred at 300 rotations per minute. There was a platinated mesh used as a counter electrode (cathode). The distance between the sample and counter electrode was ~3 cm to create a homogenous electric field around the tested surface. The anodization process took only 5 s, during which the shade of titanium substrates was significantly changed from its shiny metallic color. The anodization voltage was set to 15, 30, 45, 60, 75, and 90 Volts.

### 2.3. Surface Characterization and Color Assessment

After the anodic coloring, samples were rinsed by demineralized water and dried by warm air. Subsequently the color was assessed using a comparative method according to the RAL standard. After that, the samples were evaluated by a reflectometry method according to the ASTM D2805 method (“S2000, Ocean Optics”), when the intensity of each single reflected wavelength was evaluated separately [17,18]. The surface was observed by Scanning Electron Microscopy (SEM) FEI QUANTA FEG 450 in a back-scattered electron regime (BSE).

### 2.4. Contact Angle Measurement

The contact angle and surface energy were tested by the sessile drop method (SEE system, Advex, Ltd., Brno, Czechia) according to the ASTM D7334 standard. 2 µL droplets of high purity water were used during this experiment. The contact angle, θ, was determined by the tangent to the drop profile at the point of contact of the three phases (liquid, solid, and gas) with the plane of the sample surface [19]. Young’s Equation (1) determinates the free surface energy of the solid sample, where γSV, γLV, and γSL represent the interfacial tensions per unit length of the solid-vapor, liquid-vapor, and solid-liquid contact line, respectively [20].
γ_SV_ − γ_SL_ = γ_LV_ × cos θ(1)

### 2.5. Electrochemical Testing

After that, samples were mounted into corrosion cells, which were filled with physiological solution (0.9% NaCl, Sigma-Aldrich, Praha, Czech Republic). A 1 h time gap was used for stabilization of the corrosion reaction and corrosion potential. A Potenciostat PGZ 100 (Voltalab, Praha, Czech Repubic), equipped with Voltamaster 10 software, was used for both electrochemical testing methods. The tests were performed using a three-electrode set-up, with the sample connected as the working electrode, an extra pure carbon rod and a saturated calomel electrode (SCE) as the counter and reference electrode, respectively.

Electrochemical Impedance Spectroscopy (EIS) measurements were performed first as they are non-destructive. The amplitude of the perturbation signal was 10 mV, and the investigated frequency range was from 100 kHz to 1 Hz, with an acquisition rate of 10 points per decade.

All polarization measurements were started after another 1 h time gap. The start potential was 100 mV below the open circuit potential (OCP), the polarization rate was set to 1 mV/s, and polarization was terminated when the potential of the working electrode reached a value +500 mV vs. SCE.

### 2.6. Biocompatibility Tests

The mouse osteoblastic cell line MC3T3-E1 was obtained from the European Collection of Authenticated Cell Cultures (ECACC, Sigma-Aldrich, USA). MC3T3-E1 cells were cultured at 37 °C in a humidified atmosphere with 5% CO_2_ in cell culture alpha-Minimum Essential Medium (MEM, Biowest, Riverside, MO, USA) supplemented with 10% Fetal Bovine Serum (FBS, Biowest, USA), 100 µg.mL^−1^ of Streptomycin, and 100 U.mL^−1^ of Penicillin (Sigma-Aldrich, USA). The cells were incubated in 75 cm^2^ tissue culture flasks and subcultured when the cells reached 70–80% cell confluence. The biocompatibility of the samples was performed by extract and contact tests with MC3T3-E1 cells. The cells were detached with 0.25% (*w*/*v*) Trypsin-0.53 mM EDTA (ethylenediamine tetraacetic acid, Biowest, USA) and centrifugated at 200× *g* for 5 min. The pelleted cells were resuspended in the MEM and the density of the cells was determined with a Bürker chamber. For the extract assay, the samples were immersed in the MEM at 37 °C in the cell incubator for 72 h, at a ratio of the surface area to the volume of the extraction medium of 3 cm^2^/mL in accordance with Mohd Shafiee et al. 2021 [21]. The obtained extracts were diluted in the MEM in the concentration ranges of 10–25–50–75–100%. The MC3T3-E1 cells were seeded at 1 × 10^4^ cells per well in the 96-well cell culture plates and were cultured overnight at 37 °C in the cell incubator (Sanyo, Schoeller Instruments). The MEM was then removed and 100 µL of the tested extract was added. After the incubation of the plates for 24 h at 37 °C, the medium was aspirated from the wells, and cells were washed with Phosphate Buffer Saline (PBS, Sigma-Aldrich, USA). Then, 100 µL of the 3-(4,5-dimethylthiazolyl-2-yl)-2,5-diphenyltetrazolium bromide (MTT, Sigma-Aldrich, USA) solution (0.5 mg.mL^−1^) was added to each well and the plates were incubated in the cell incubator for 2 h. The yellow water-soluble MTT substrate is converted by the metabolically active cells to a water-insoluble purple formazan. The solutions in the wells were removed and 100 µL of the dimethylsulfoxide (DMSO, VWR International, France) was added to each well for dissolving the formazan. The plates were shaken at room temperature for 20 min. Finally, the amount of purple formazan produced by metabolically active cells was quantified at a wavelength of 570 nm using a spectrophotometer (Epoch, BioTek, Winooski, VT, USA). The results were assessed as the percentage of viability compared to the control (untreated cells) which was considered 100% viable [21]. The samples were tested in six parallels and data were expressed as the mean ± standard deviation (SD). Statistical analysis was determined using one-way ANOVA with Tukey’s multiple comparison test. All statistical analyses were executed using the program R (R Core Team, Vienna, Austria).

## 3. Results

### 3.1. Surface Character and Color Assessment

After the anodic coloring and cleaning process, the colors of samples were determined by the RAL standard, which is typically used for the color coding of medical tools and equipment [2]. The color palette produced by anodic coloring is illustrated in Figure 1. The RAL color codes for each anodization voltage are listed in Table 1.

The surfaces were tested for reflectance characteristics. Only wavelengths of visible light were used for testing. The varying intensity of reflected light for single wavelengths was recorded. The graphical interpretation of reflectance specters for each anodic coloring voltage is presented in Figure 2, where the colors of curves represent the color of the surfaces.

The morphology changes caused by anodization process were studied by SEM methods. The major variation was observed in BSE mode, when the area of anodic coloring appears darker due to formation of titanium oxide layer based on TiO_2_ on the surface [4,22]. Figure 3 illustrates the interface between an anodized and non-anodized surface and indicates that there was no significant difference in surface topography found on the anodized surface.

### 3.2. Contact Angle Measurement

There were 10–12 droplets analysed on each sample as only limited parts of surfaces were eligible for this test; best results are obtained when droplets are analysed near to the edge of the sample. The surface energy of each coloured specimen was calculated by the Li-Neumann method [23]. Mean values and standard deviations are listed for each anodization voltage separately in Table 2.

### 3.3. Electrochemical Testing

After a 1 h immersion in isotonic physiological solution, the EIS measurement was performed. The values of real impedance (*Zr*) and imaginary impedance (*Zi*) were obtained for each frequency. These values were later used for the calculation of absolute impedance (*Z*), according to Equation (2), and phase shift (*θ*), according to Equation (3) [24].
(2)$$Z=\sqrt{Z_{i}{}^{2}+Z_{r}{}^{2}}$$
(3)$$\theta =arc\;tg\;\left(\frac{Z_{i}}{Z_{r}}\right)$$

Bode (Figure 4) and Nyquist (Figure 5) diagrams were assembled from values obtained by EIS and further calculations according to equations above. These diagrams truly represent electrochemical behavior of different surface states under an aerated physiological solution environment. The colors of the curves match the colors of the anodized surfaces. The equivalent circuit shown in Figure 5 was found to represent the electrochemical elements of the anodized layer and testing solution properties.

EIS specters were fitted with an equivalent circuit by EIS Spectra Analyser 1.0 and parameters of each element were obtained. Parameter values for different anodization voltages are listed in Table 3. Resistance of all the circuit connectors and corrosion solution itself are summarized as a value of “Resistance solution” (R solution) [25]. Resistance of the layer (*R layer*) and parameters (*P*, *n*) of the constant phase element (CPE) were used for the calculation of anodized layer capacitance (*C layer*) according to the Equation (4) [26].
(4)$$C_{layer}=\frac{{\left(P*R_{layer}\right)}^{\frac{1}{n}}}{R_{layer}}$$

One polarization curve was measured for every anodically coloured surface. The relation of corrosion current on polarization potential was recorded during the polarization procedure. Common corrosion parameters were determined by Tafel extrapolation [27] on the curves. Polarization curves are presented as a semilogarithmic axial system to illustrate the value of corrosion potentials. This was done in a typical “V-like” shaped area of the curve, where anodic reactions begin to predominate over cathodic ones. The measurement procedure was automatically terminated when either potential reached 500 mV SCE. The relation of corrosion current on polarization potential is illustrated in Figure 6.

The corrosion parameters were determined automatically from the so-called Tafel region by Voltamaster 10 software and were accepted only if correlation deviation did not exceed 0.5%. The corrosion rate was further calculated from corrosion current density using the Farraday equation [28], where density equaled 4.5 g/cm^3^ and the valency of corroded ions 4 were considered for calculation. Corrosion parameters of tested samples are listed in Table 4.

### 3.4. Biocompatibility and Cytotoxicity

MTT assay was used to evaluate the cytotoxicity of the anodic colored samples. The sample is considered potentially cytotoxic if cell viability falls below 70% of the control viability (untreated cells) [29]. Table 5 shows the percentage of MC3T3-E1 cell viability calculated from the absorbance measured at 570 nm. The viability of the control cells was considered as 100%. The cytotoxic effect on MC3T3-E1 cells was not detected in any tested titanium material, neither by the extract and contact test. No significant difference was observed between the 15 V, 30 V, 45 V, 60 V, and 75 V materials (*p* > 0.05).

## 4. Discussion

The coloring of titanium by anodization is produced by means of the intervening action of the reflecting light from the titanium samples surface covered with mixture of titanium rich oxides and hydroxides film and the reflecting light of interface between this film and matrix of titanium substrate [13]. When titanium substrate is immersed in solution and an electric current passed through, oxygen is produced on the anodized substrate surface. Molecules of oxygen intermediately bond with titanium atoms forming oxides, whose structure is primarily determined by the applied voltage and the activity/chemical composition of solution. Different microstructures and thicknesses of oxide film causes variation of refractive and reflective indexes, or luminous flux, which results in changes of shades or colors of anodized surfaces [30]. There were various compositions of anodizing solutions pre-tested before this research. It was found that solutions containing organic acids produce more pronounced and brighter colors. Colored layers are also more compact with better adhesive connection to the substrates. When oxidative inorganic acids were used in solution composition, there were blisters, layers delamination, and other smaller or lager imperfections observed in the emerging-colored layers. Therefore, a solution of citric acid was used for this experiment.

A reflectometry method was used for recording the reflected specters of light, which differed for each wavelength. As shown in Figure 2, there is an obvious shift of maximal values of reflected light to higher wavelengths for higher applied voltage. According to that, there is an assumption that surfaces anodized at higher voltage preferably absorb light radiation of longer wavelengths, but on the other hand, lighter of shorter wavelengths is reflected [31]. Unfortunately, the curve of samples anodized at 75 V does not correspond to this affirmation – its minimum value of reflectance occurs at ~460 nm, which is similar also for 15 V anodization. According to previous research there is another assumption that says increasing anodization voltage produces periodically a repetition of colors, but their shade becomes brighter and more saturated at higher voltages [32]. It was also found that color is not determined by the time of anodization, but only by applied voltage, which means that colors are not produced by light intervention, but by the absorption of particular wavelengths from the light spectrum. It was confirmed that very long anodization times result in surface blistering and deadhesion of the anodized layer [10].

Titanium surface coloring became very popular especially for the marking of implants for easier and reliable differentiation. For better orientation, each implant set is accompanied by a list of sizes, and each size is marked with different colors. Therefore, it is so important to find which anodization voltage produces the required colors. These colors can be found at the RAL list of standards under its specific name and code. This standard is used internationally and is an European alternative to the ASTM standardized list of colors [33]. As illustrated in Figure 1, 15 V and 75 V anodization produced yellow/brown colors shades and 30–60 V anodization resulted mostly in blueish shades. These results also correspond to previous studies [4,31,34].

There were only statistically irrelevant differences of wettability found between all colored samples. The mean value of contact angles for all surfaces was ~ 81°. The wetting angle of untreated surfaces titanium gr. 4 referred to in [35,36,37] was between 90–110° As all the surfaces were shiny and mirror like, the effect of roughness was not considered into the calculation. The value of standard deviation was also insignificant as the surface coloring was very homogenous without any macro or microdefects, and the contact angle was very similar for all evaluated droplets. As published earlier [38], fibrose cells preferably bond to implants and grow on implants surface the best if the contact angle of the exposed surface is 60–80°. The wettability of tested samples predetermines titanium with colored surfaces for the manufacturing of short- and medium-term implants, as there is no need to not bond with the bone strictly. There is a prediction of fibrous cells layer growing of the surface of colored titanium implants which allow them to be easily removed after their proper time of service. For example, internal bones screws or fixators are commonly removed when damaged bone is restored completely [39]. If the contact angle will be significantly lower (close to zero value), growth of bone cells will be preferred on the surface which would make the titanium implants difficult to remove and there is high probability of damage to surrounding tissue during the process of its removal [40]. The surface energy of the implant surface obtained by the Li-Neumann method [41] has a significant impact on the tissue integration process. The mean value of free surface energy, obtained by testing using water of high purity, was ~34.5 mJ/m^2^.

The effect of anodization on electrochemical properties was studied by electroimpedance spectroscopy to find curves of electrical impendence. Characteristic impedance curves presented in Figure 4, Figure 5 and Figure 6 show the impedance shift for each anodization voltage. There is a significant difference between 75 V and the rest of curves. There is an assumption that anodization at high voltage produces very compact surface layers with highly organized titanium oxide in its microstructure. This layer contains less microdefects and behaves like a highly effective electric insulator [42]. As there is large amount of oxygen molecules developed in anodization process at higher voltage, there in not enough time and driving force to react with titanium atoms and form into a deffectless structure. Therefore, the samples anodized at higher voltage show a different shape of impedance curve as the colored layer on their surface probably contains high amount of morphologic defects [43]. The summarized resistance of the solution was very close for all samples, signaling a similarity to the constant testing conditions. The resistance of the anodized layers obtained by EIS showed the same trend as resistance obtained by potentiodynamic polarization. The difference between the two values is probably caused by the dynamics of electric charge transfer across the anodized layer [44]. The values of capacitance of the anodized layers are independent of the voltage of anodization, signaling that the layers show similar physical and electrochemical properties [45]. Surfaces anodized at 75 V show significantly lower values for both the layers’ resistance and capacitance. This phenomenon might be caused by microscopic defects formed due to high oxygen evolution during the anodization process at high voltage, resulting in the formation of cavities, microcracks, or even microblistering [14]. The insulating parameters of the defective layer are therefore decreased, resulting in lower resistance [46] and layer capacitance [11].

Potentiodynamic polarization in physiological isotonic solution proved there are differences between corrosion properties of samples anodized at different voltages. There was significant relation between anodization voltage and corrosion potential observed. It was found that higher coloring voltage produced surface layers with more noble corrosion potential, −356 mV (15 V) vs. −193 mV (75 V). The values of polarization resistance of surface layers produced at higher voltage is also higher. The lower value of polarization resistance of surface anodized at 75 V indicates the presence of defects in studied surfaces which was previously confirmed by the EIS method [11]. A more noble corrosion potential of surfaces decreases the risk of bimetallic corrosion if combined directly with another material. Higher polarization resistance finally results in a lower corrosion rate if corrosion damage occurs in the case of bad implant design or a highly aggressive environment. A more noble corrosion potential also shows that higher voltage creates a more stable oxide layer on the surface [47,48]. Coloring anodization can be evaluated as highly positive with regard to the corrosion rate of the samples. Even maximal corrosion rate for implants considered as “limit” is 0.13 mm/year, colored samples showed significantly lower corrosion rate [49]. A potentiodynamic test proved that surfaces colored at 15 V will corrode 2.1 nm/year and the 75 V sample will corrode as little as 0.7 nm/year. This is significantly below the corrosion rate limit for implants and prosthesis. The corrosion rate of untreated titanium gr. 4 was previously referred to as 10–500 nm/year depending on surface treatment (fresh grinding, acid etching, etc.) [12,50,51]. According to these results, titanium coloring may be highly recommended as a post-manufacturing surface treatment to increase final application corrosion properties [52,53].

Medical devices have been widely used in various clinical disciplines. As these devices have direct contact with the tissues and cells of the body, they not only require good physical and chemical properties, but must also have good biocompatibility [54]. The cytotoxicity test, one of the biological evaluation and screening tests, uses tissue cells in vitro to observe the cell growth, reproduction, and morphological effects by the medical devices. Cytotoxicity is one of the most important methods for biological evaluation as it has a series of advantages, along with the preferred and mandatory items [55]. The level of viable cells after a 48-hour direct contact test was similar for all samples anodically colored at different voltages and exceeded 99%. Other studies performed on titanium of comparable grades also informs about 95–99% fibroblast viability [56,57]. There were some cytotoxicity problems found in the case of titanium alloys where the release of vanadium and aluminum decreased cell viability significantly, especially after long term contact tests [54,56,58]. Therefore, highly alloyed titanium materials are unsuitable for longer term use as dental implants, where the release of the ions may be accelerated by the aggressive environment [59]. On the other hand, the pure titanium after additional surface treatment, like presented in this study, shows only insignificant ion release level connected to the high cells viability during cytotoxicity test [60].

## 5. Conclusions

The anodic coloring was performed on CP titanium grade 4 in citric acid solution. The contact angle tests results indicate the preferable use of this technique for color coding of implants for short and mid-term use. The corrosion rate obtained by potentiodynamic polarization was very low, indicating only insignificant general corrosion of the surface. Impedance specters of colored surfaces indicate the presence of highly non-conductive layers protecting the surfaces against the influence of surrounding environment. Anodically colored surfaces showed very high biocompatibility and nearly no cytotoxicity. Results of presented paper should be further extended by studies focused on long-term corrosion properties of titanium color coded surfaces and the effect of the process should be furthermore evaluated in the context of bacterial adhesion.

## Figures and Tables

**Figure 1 materials-16-00632-f001:**
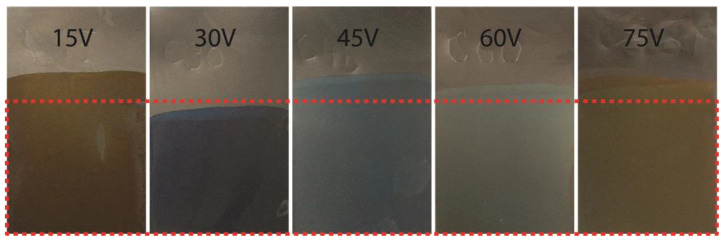
Color palette produced by anodic coloring at different voltages.

**Figure 2 materials-16-00632-f002:**
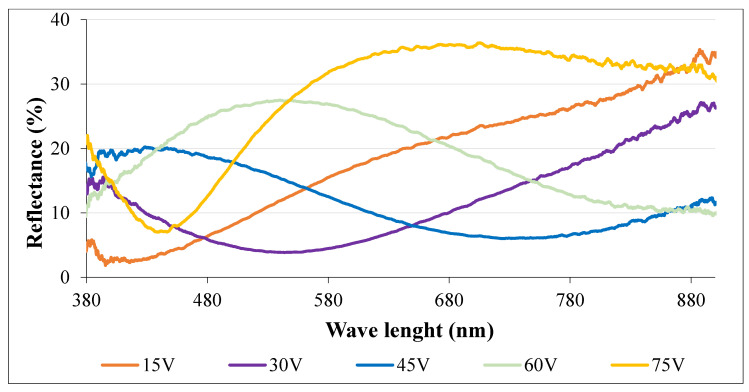
Reflectance specters for samples anodized at different voltages.

**Figure 3 materials-16-00632-f003:**
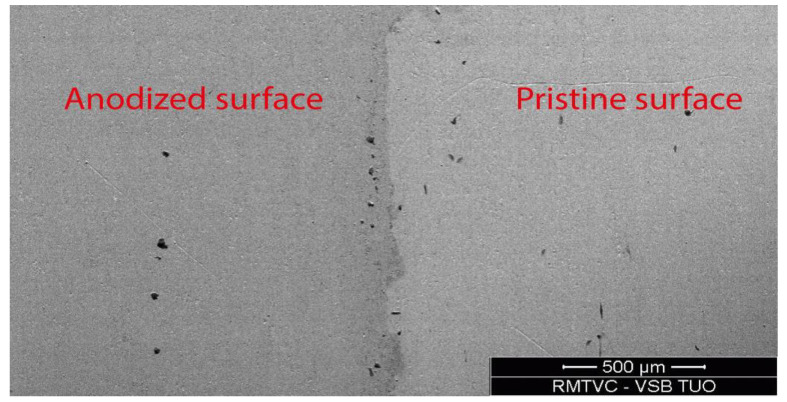
Comparation of an anodically colored and a pristine surface (SEM-BSE).

**Figure 4 materials-16-00632-f004:**
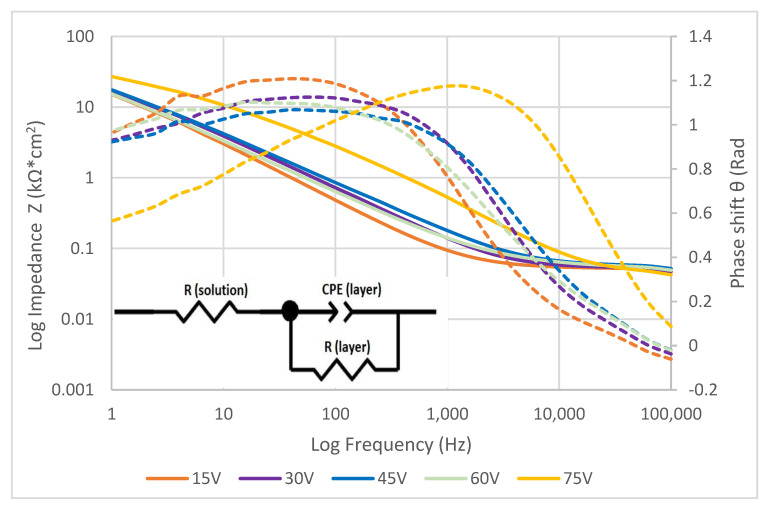
Bode diagram for anodized surfaces (full line for Impedance, dotted line for Phase shift).

**Figure 5 materials-16-00632-f005:**
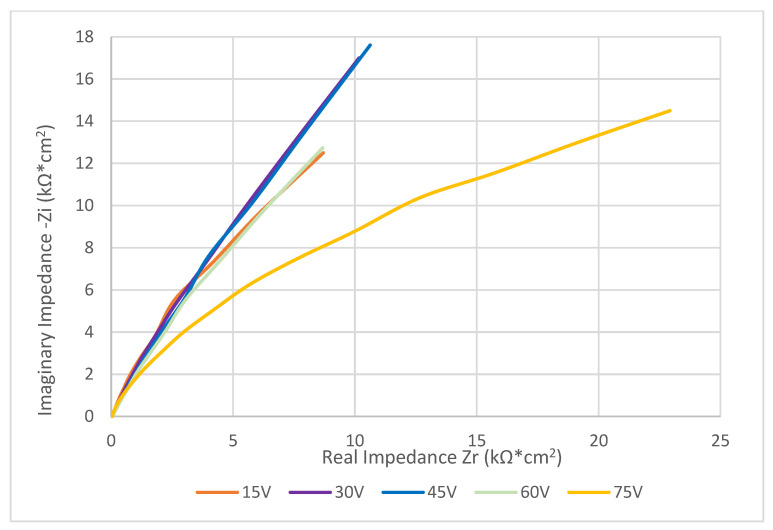
Nyquist diagram for surfaces anodized at different voltages.

**Figure 6 materials-16-00632-f006:**
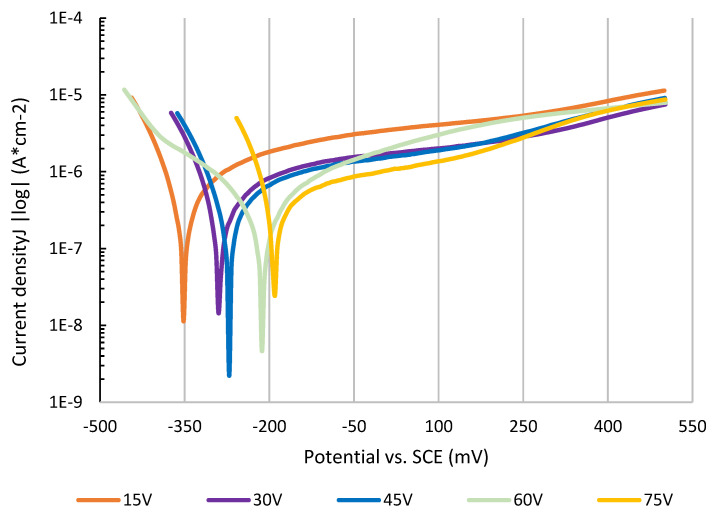
Polarization curves for anodized surfaces.

**Table 1 materials-16-00632-t001:** RAL codes for colors produced by each anodization voltage.

Voltage (V)	RAL (Name)	RAL (Code)
15	Clay brown	8003
30	Steel blue	5011
45	Sky blue	5015
60	Pale green	6019
75	Ochre yellow	1024

**Table 2 materials-16-00632-t002:** Contact angle of anodically colored surfaces.

Anodization Voltage (V)	Contact Angle (°)	Surface Energy (mJ/m^2^)
15	81 ± 2	35.0
30	82 ± 4	34.1
45	80 ± 2	35.3
60	81 ± 4	34.8
75	82 ± 3	34.1

**Table 3 materials-16-00632-t003:** Values of elements of fitted equivalent circuit.

Anodization Voltage (V)	Resistance of Solution (Ω)	Resistance of Layer (Ω·cm^2^)	P	n	Capacitance of Layer (F·cm^2^)
15	48.59	4.32 × 10^4^	1.17 × 10^−5^	0.81	1.00 × 10^−5^
30	48.411	5.08 × 10^4^	1.12 × 10^−5^	0.76	9.33 × 10^−6^
45	51.581	6.12 × 10^4^	1.19 × 10^−5^	0.72	1.05 × 10^−5^
60	50.435	6.93 × 10^4^	1.44 × 10^−5^	0.74	1.44 × 10^−5^
75	36.086	2.93 × 10^4^	2.54 × 10^−6^	0.76	1.11 × 10^−6^

**Table 4 materials-16-00632-t004:** Corrosion parameters of anodically colored surfaces.

Voltage (V)	E_corr_ vs. SCE (mV)	Corr. Rate (nm/year)	Polar. Resistance (kΩ·cm^2^)
15	−356	2.1	63
30	−278	1.1	79
45	−256	1.5	85
60	−221	1	82
75	−194	0.7	35

**Table 5 materials-16-00632-t005:** Cell viability after a contact and extract test was performed on the anodic colored surfaces.

Cell Viability (%)		
Sample	Extract Test	Contact Test
Ti-15V	99.7 ± 4.3	99.2 ± 3.1
Ti-30V	99.2 ± 3.8	99.5 ± 3.6
Ti-45V	100.0 ± 0.9	99.4 ± 3.1
Ti-50V	100.0 ± 2.1	99.7 ± 4.0
Ti-75V	99.5 ± 3.3	99.5 ± 3.9

## Data Availability

There are no data other than presented in the manuscript.

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
