# Peer review of "Electrochemical, Biological, and Technological Properties of Anodized Titanium for Color Coded Implants"

_materials, 2023, doi:10.3390/ma16020632_

Round 1

Reviewer 1 Report

This paper is about "Electrochemical, biological and technological properties of ano- 2 dized titanium for color coded implants" and Commercial titanium grade 4 plates 50 x 20 x 0.1mm 19 were tested to obtain their electrochemical and other technological properties.

There are minor revision issues with this paper.

1- The supriority of this method from the other methods in the literature must be written in the abstract part of the manuscript methods.

2- Introduction part can be developed with more references.

There are many places need rewriting

a. Line 19.... "CP titanium" must be written as "Commercial titanium"

b. Line 45 there is a blank after nucleate before [3]

c. Line 52 "therefor"  must be written as "therefore"

d. Line 64 there is no dot after "bath"

e. Line 66 "C6H8O7" must be written as "C6H8O7"

f. Line 70 "RPM" must be written as "rpm"

g. LÄ°ne 110 "CO2" must be written as "CO2"

h. Line 121 "3 cm2" must be written as "3 cm2"

i. Line 260 "Therefor" must be written "Therefore"

j. Line 294 "Therefor" must be written "Therefore"

Author Response

Dear reviewer.

Many thanks for your time spent on this revision. There was a part dedicated to the biological and clinical significance of surface anodization added to the introduction. Further research direction was also added to the conclusion -we already started the long-term corrosion tests of anodized surfaces and even the tests of the stability of the layer during simulated implantation (artificial bone) were started and will be published later next year.

All (hopefully) the typos, misspells and text imperfections were double-checked and corrected properly. The mention about the significance of the used anodization solution was stated in the abstract and more details about it were added to the introduction.

I‘m sure that the text added to the manuscript on your recommendations will improve its quality and it will become more interesting for the readers.

Many thanks.

Reviewer 2 Report

The paper by Hlinka et al provides an interesting study on the properties of anodized titanium that could have an high impact if the author could suggest the clinical and biological significance of the characteristics investigated.

In my opinion the introduction section is missing of a biological background on the role of physical aspects and biological implications that the authors could find in 

"Lauritano D, Moreo G, Lucchese A, Viganoni C, Limongelli L, Carinci F. The Impact of Implant-Abutment Connection on Clinical Outcomes and Microbial Colonization: A Narrative Review. Materials (Basel). 2020 Mar 3;13(5):1131. doi: 10.3390/ma13051131. PMID: 32138368; PMCID: PMC7085009."

In the conclusion section the authors should clarify what are the main directions that further studies could follow.

Author Response

Dear reviewer.

Many thanks for your time spent on this revision. There was a part dedicated to the biological and clinical significance of surface anodization added to the introduction. For that I've used the paper you suggested.

Further research direction was also added to the conclusion -we already started the long-term corrosion tests of anodized surfaces and even the tests of the stability of the layer during simulated implantation (artificial bone) were started and will be published later next year.

I‘m sure that the text added to the manuscript on your recommendations will improve its quality and it will become more interesting for the readers.

Many thanks

Reviewer 3 Report

This manuscript, entitled „ Electrochemical, biological and technological properties of anodized titanium for color coded implants” is relevant to the scope of this journal.

It is an interesting study that can provide valuable information to specialists in the field.

The authors made a good synthesis of the literature that provides an overview of the research evolution in this area.

Therefore, the article can be recommended for publication only after mandatory revision according to the following suggestions:

 1. Pay attention to small typos like the one in line 64 "bath. no further surface" or line 94 "PGZ 100 potentiometer".

2. Consistency in writing style should be maintained throughout the manuscript. For example, the sign to multiply is "." or "*".

3. The chemical composition of the studied material must be presented.

4. Equations 2 and 3 do not need to be specified because they are very well known to electrochemists.

5. Since this anodizing technique is intended to be used to improve the anticorrosive performance of titanium, the non-anodized surface must also be presented as a reference, both in electrochemical and biological studies or measuring the contact angle value. If there are no determinations made by the authors, at least references in the chosen corrosive environment can be found.

6. Figure 4 does not show the Bode diagram. It's just part of it. In the complete Bode diagram, the impedance module is represented in a logarithmic scale depending on the frequency, also a logarithmic scale, superimposed on the phase angle (in degrees) depending on the frequency (logarithmic scale). Therefore, Figures 4 and 6 must be joined and modified, making the necessary transformations. And a point representation of experimental data would be more efficient.

7. The results obtained from the EIS tests need to be better explained. In order to have a justification, the data should be fitted with electrical equivalent circuits, and the parameters obtained from these fittings should be presented and discussed accordingly.

8. Please check the data presented in Table 3 and the polarization curves in Figure 7. From this figure, it seems that the sample anodized at 45V has the lowest corrosion density.

Author Response

Dear reviewer.

Many thanks for your time spent on this revision.

We have done our best with the final changes according to your suggestions.

Please allow me to add a short comment to each of your points:

1) The typos were checked by nearly all authors and corrected properly.

2) The different signs were corrected and are (hopefully) constant for the manuscript.

3) The chemical composition of the material was added to the text and also the ASTM standard was mentioned.

4) I’d like to retain the equations in the text due to possible “non experienced” readers. But if you insist on their removal, I’ll delete them.

5) The result for the pristine surface was added in form of citations as it was not measured within this research.

6) Bode diagram was rearranged-Thank you for this point, I had always presented the phase shift and Z/f relation separately, but it looks better this way!

7) There was an equivalent circuit found and parameters were fitted. The values of each element are listed in a separate table, and they are also properly discussed.

8) The corrosion rate value is OK in this case-the “V” shape is “deeper” for 45V, but if you look at the direction of the anodic part of the curve, you can see the intersection between the cathodic and the anodic part of the curve is slightly higher than for 75V. I’ve checked the original date from the potentiostat, and they are correct.

I’m sure that changes done on your behalf will improve the quality of the paper it will become more interesting for the readers.

Thank you

Josef Hlinka

Round 2

Reviewer 3 Report

The authors have made all the requested corrections and additions, so that the manuscript now has a much improved form that can be considered for publication.